# Multivariate Dense Retrieval: A Reproducibility Study

## Abstract

The current paradigm in dense retrieval is to represent queries and passages as low-dimensional real-valued vectors using neural language models, and then compute query-passage similarity as the dot product of these vector representations. A limitation of this approach is that these learned representations cannot capture or express uncertainty. At the same time, information retrieval over large corpora contains several sources of uncertainty, such as misspelled or ambiguous text. Consequently, retrieval methods that incorporate uncertainty estimation are more likely to generalize well to such data distribution shifts. The multivariate representation learning (MRL) framework proposed by Zamani & Bendersky (2023) is the first method that works in the direction of modeling uncertainty in dense retrieval. This framework represents queries and passages as multivariate normal distributions, and computes query-passage similarity as the negative Kullback-Leibler (KL) divergence between these distributions. Furthermore, MRL formulates KL divergence as a dot product, allowing for efficient first-stage retrieval using standard maximum inner product search.

In this paper, we attempt to reproduce the MRL framework for dense retrieval by Zamani & Bendersky (2023). We find that the original work (i) introduces a typographical/mathematical error early in the formulation of the method that propagates to the rest of the original paper's mathematical formulations, (ii) does not provide all of the necessary information to facilitate reproducibility, and (iii) proposes a training setup to train MRL that, if followed, does not yield the reported performance in a fair comparison. In light of the aforementioned, we address the mathematical error, make some reasonable design choices, and propose an improved training setup that complements the original paper by filling in important details that were unspecified. We further contribute a thorough ablation study which is absent from the original paper, to gain more insight into the impact of the framework's different components. Despite our efforts, we were neither able to reproduce the exact results reported in the original paper, nor to uncover the reported trends against the baselines. Our analysis offers insights as to why that is the case. Most importantly, our empirical results suggest that the definition of variance in MRL does not consistently capture uncertainty. The source code for our reproducibility study is available at: `https://anonymous.4open.science/r/multivariate_ir_code_release-AB26`.

## 1 Introduction

Dense retrieval has become the new paradigm in first-stage retrieval, largely replacing lexical methods which cannot model semantic information as well as neural models. Dense retrievers following the dual-encoder architecture (Karpukhin et al., 2020) are popular first-stage retrievers due to their performance and scalability. This paradigm leverages pre-trained neural language models to encode queries and passages as low-dimensional real-valued dense vectors, with relevance defined as their dot product. Passages are encoded offline and stored in a dense index. At query time, retrieval can be done efficiently using maximum inner product search (MIPS). However, representing queries and passages as single vectors has an important limitation that has influenced the research landscape: these representations do not model, capture, or express predictive uncertainty or risk. At the same time, there are various sources of uncertainty arising both from the data and the neural retrieval models:

**Query uncertainty.** User queries may include misspellings, ambiguity, and incomplete or inaccurate information (e.g., false memories). Furthermore, in a realistic setting, the retrieval system has minimal to no prior knowledge about the distribution of the queries, except for possibly a few assumptions (e.g., the language they are in, or common user mistakes).

**Passage uncertainty.** Passages may present similar uncertainty-inducing artifacts to queries, such as misspellings and ambiguity. Unlike queries, the retrieval model has prior knowledge of a passage collection.

**Relevance uncertainty.** Relevance, or ranking uncertainty refers to the confidence of the model in the estimated query-passage relevance. Such an estimator may be anything between a deterministic function of query and passage uncertainty e.g., the model reproduced in this paper, to a stochastic function of deterministic query and passage representations e.g., a Monte-Carlo dropout Bayesian estimator (Cohen et al., 2021).

Uncertainty estimation remains largely unexplored for the case of first-stage dense retrieval, despite it having received increased attention from the community in the case of re-ranking (Wang & Zhu, 2009; Zhu et al., 2009; Feng et al., 2020; Cohen et al., 2021; Heuss et al., 2023). Recently, Zamani & Bendersky (2023) proposed the multivariate representation learning (MRL) framework, the first approach that models uncertainty in the context of dense retrieval. MRL uses predictive variance as a proxy for uncertainty. Each query and passage is mapped to a multivariate Gaussian distribution parameterized by a mean vector and a (diagonal) covariance matrix, where the mean represents the predicted query or passage embedding, and the variance represents the uncertainty of said embedding. However, different from existing approaches to modelling uncertainty in IR that leverage Bayesian inference (Cohen et al., 2021), MRL treats both the mean and the variance as point estimates. In that sense, variance in the MRL framework does not express statistical variance, i.e., deviation from the mean, as much as it expresses predicted risk. In essence, this is a trade-off between being theoretically principled and computationally efficient. Admittedly, computational efficiency is of the utmost importance in first-stage retrieval, where one has to manage collections of potentially billions of passages.

Having represented queries and passages as multivariate normal distributions, the authors of the original paper proceed to formulate a query-passage relevance scoring function based on a simplified version of the Kullback–Leibler (KL) divergence. Further, they express this function as a dot product between query and passage representations, thereby allowing for efficient retrieval by means of standard MIPS (e.g., FAISS (Johnson et al., 2021)). Finally, they report state-of-the-art retrieval performance, and show that the predicted covariance matrix could be used as a pre-retrieval query performance predictor.

Even though the results reported in the original study showcase the effectiveness of the proposed method, our study is motivated by several important questions that still need to be explored. First and foremost, the fact that neither the source code nor model checkpoints are released, makes it hard to verify the paper's substantial claims. Second, even though the model consists of various components and several stages of knowledge distillation, the original work does not include an extensive ablation study that explores the impact of each on downstream performance. Therefore, whether the performance gains come from representing the queries and passages as distributions is unclear. To this extent, it is important to understand the representations learned by MRL. In this reproducibility work, we aim to answer the following research questions:

**RQ1 Reproducibility:** Can the models proposed in Zamani & Bendersky (2023) be reproduced by following the methodology and experimental setup outlined in the original paper?

**RQ2 Analysis:** Can the multivariate query and passage representations express uncertainty?

**RQ3 Ablation Study:** What is the contribution of each component that comprises MRL to downstream retrieval performance?

Furthermore, we summarize our contributions to the original work:

**Correction of a typographical/mathematical error.** We correct a typographical/mathematical error of the original work, made early in the formulation of the method, in an attempt by the authors to simplify the computation of KL divergence between two multivariate normal distributions. This error propagated to

the rest of the original paper's mathematical formulations. We hypothesize that this error did not further propagate to the implementation of their experiments, as we provide empirical evidence that if the incorrect similarity function is used instead of our corrected version, it harms retrieval performance.

**Reproduction of the MRL model.** We attempt to be as faithful as possible to the reported training regime. Due to a few details omitted in the original work, we test and validate several plausible variants of the proposed method. We show that some training setups result in much lower effectiveness than others, and highlight effective strategies that result in a competitive retrieval model. We contextualize each strategy in the context of the original paper and justify these choices.

**Reproducing retrieval and QPP experiments.** We reproduce the experimental setup of the original paper, for the tasks of dense retrieval and pre-retrieval query performance prediction (QPP). We were not able to obtain the exact results reported in the original study or confirm the original findings using our experimental setup. We show that even though MRL is a highly competitive approach, it does not outperform the baselines in a fair comparison (e.g., MRL does not outperform baseline method when the same batch size is utilized across baselines) Additionally, MRL yields inconsistent results for the QPP experiments, with our analysis revealing that the variance vectors do not consistently capture notions of uncertainty.

**Ablation study.** The MRL framework is composed of multiple components, including multivariate representations, knowledge distillation, and model initialization from an already effective pre-trained dense retriever. To unveil the importance of each component in training an effective retriever, we conduct a thorough ablation study. For example, how much of the reported performance increase can be attributed to the core contribution of the paper, which is the use of multivariate representations? Our elaborate ablation study uncovers the impact of each component on the retrieval performance. In particular, we find that multivariate representations do not boost the model's performance, and the high effectiveness stems from the model initialization and knowledge distillation from the re-ranker.

**Proposed improvements upon the original MRL model.** We propose a simple alteration to the original model, which results in a reduced hyperparameters search space. In short, instead of a parametric softplus activation which ensures positive semi-definiteness of the covariance matrix, we propose predicting the log-variance instead, which obviates searching for the $\beta$ hyperparameter of the softplus function. We show that the log-variance model either matches or outperforms the original softplus model.

## 2 Related Work

**Uncertainty-aware retrieval.** Uncertainty estimation in neural information retrieval (IR) has been explored in the past, although not in the context of dense (first-stage) passage retrieval, which is the main novelty aspect of the MRL method. The work of Cohen et al. (2021) and Heuss et al. (2023) focuses on risk-aware (second-stage) re-ranking. Both approaches attempt to approximate Bayesian models that predict a distribution of relevance scores rather than point estimates; the former utilizes Monte-Carlo dropout (Gal & Ghahramani, 2016), while the latter leverages Laplace approximation. The common denominator across these Bayesian methods is that the predictive distribution $p(y|\theta, \mathcal{D})$ is approximated by performing forward inference using multiple samples of $\theta$. This is the main difference between prior work and MRL: In MRL, predictive uncertainty is not framed as weight uncertainty, and variance does not represent deviation from the mean prediction. Rather, variance in MRL is a predicted value of a deterministic estimator.

**Uncertainty for detecting out-of-distribution/corruptions.** While a variety of efforts exist in the area of stochastic representations in image retrieval (Warburg et al., 2021; Chun et al., 2021), recent work by Warburg et al. (2023) showed that Bayesian image retrieval with Laplace approximation can achieve some desirable properties. They show that the uncertainty of prediction increases (almost monotonically) with the amount of corruptions in the input. The model's predictive uncertainty further behaves as expected when making out-of-domain predictions. These insights are valuable in text retrieval as well, where these desirable properties have not yet been achieved effectively.

**Knowledge distillation.** The MRL framework is surrounded by multiple layers of knowledge/parameter distillation, which we summarize in this section. First, the selected neural architecture employed in MRL is DistilBERT (Sanh et al., 2019); a distilled version of BERT with 40% fewer parameters for 97% of its

original performance. Furthermore, the architecture has been distilled with balanced topic-aware sampling (TAS-B) (Hofstätter et al., 2021), that uses two teacher models to construct better training batches. Finally, MRL itself utilizes a knowledge distillation loss inspired by CLDRD (Zeng et al., 2022). While the original paper does not discuss how these sources of distilled knowledge affect downstream performance, in Section 4 of this paper we perform a thorough ablation study that examines them one-by-one.

## 3 Methodology

The proposed MRL framework represents queries and passages as multivariate Gaussian distributions. It does so by computing a mean vector $\mu$ and a diagonal covariance matrix $\Sigma$ for a query $q$ and passage $d$, using query and passage encoders $f_\theta$ and $f_\phi$, parameterized by $\theta$ and $\phi$ respectively,

$$\mu_Q, \Sigma_Q = f_\theta(q), \tag{1}$$
$$\mu_D, \Sigma_D = f_\phi(d). \tag{2}$$

It is also possible to have $\theta = \phi$, i.e., weight sharing, which the authors of the original paper opt for. Dense retrieval models (e.g., DPR Karpukhin et al. (2020)) typically utilize the embedding of a special token, the `[CLS]` token as the low-dimensional representation of queries and documents. Given a piece of text, for instance, 'Hello world', pre-processing appends the special `[CLS]` token to the start of the text, producing '`[CLS]` Hello world', and the output of the transformer model for the `[CLS]` token is used. Relevance is a function of representations of query and document, typically a dot product or cosine similarity. MRL however produces two vectors per input, which motivates the choice in the original study to use an additional special token, termed the `[VAR]` token, appended after the `[CLS]` token, but before the text. For instance, 'Hello world' is pre-processed to '`[CLS]` `[VAR]` Hello world'. The output representation of the `[CLS]` token is used to compute the mean, and the output representation of the `[VAR]` token is used to compute the variance.

The relevance score between queries and passages is then defined as the negative KL divergence between their distributional embeddings: $Q \sim \mathcal{N}(\mu_Q, \Sigma_Q)$ and $D \sim \mathcal{N}(\mu_D, \Sigma_D)$,

$$\mathrm{rel}(q, d) = -\mathrm{KLD}(Q\|D). \tag{3}$$

The minus sign is there to implement a "higher is better" type of scoring. To simplify matters, we will disregard it in the upcoming derivations and re-introduce it at the very end.

In this section, we detail the reproducibility study of the above framework. First, we direct attention to a small mathematical error that was made early in the formulation of the relevance scoring in the original paper, that propagated through the rest of the mathematical derivations. We then discuss matters of model training. Whenever we make a strong assumption due to the lack of implementation detail in the original paper, or the lack of shared source code, it is explicitly mentioned.

### 3.1 KL divergence-based relevance scoring

In Eq. 4, we start by repeating the standard definition of KL divergence, as written in Eq. 9 of the original paper:

$$\mathrm{KLD}(Q\|D) = \frac{1}{2}\Big[\log\frac{\det\Sigma_D}{\det\Sigma_Q} - k + \mathrm{tr}\{\Sigma_D^{-1}\Sigma_Q\} + (\mu_Q - \mu_D)^{\intercal}\Sigma_D^{-1}(\mu_Q - \mu_D)\Big], \tag{4}$$

where $k$ denotes the dimensionality of the multivariate Gaussian embedding. For the purpose of relevance scoring, the authors proceed to further simplify Eq. 4 and reformulate it as document ranking function. They do so by eliminating document-independent terms and constants, and by taking advantage of the fact that the covariance matrices are diagonal. Let us follow their simplification steps by considering each term separately. For the first term we have,

$$\log\frac{\det\Sigma_D}{\det\Sigma_Q} = \log\det\Sigma_D - \underbrace{\log\det\Sigma_Q}_{\substack{\text{constant w.r.t.}\\\text{doc. ranking}}} = \log\det\Sigma_D = \log\prod_{i=1}^{k}\sigma_{i_D}^2 = \sum_{i=1}^{k}\log\sigma_{i_D}^2. \tag{5}$$

The subsequent steps in the original paper contains a error in the simplification of the second term. We include the original formulation in Appendix A for completeness. We note that using the original formulation leads to drastically lower performance, which makes it likely that this error is typographical i.e., it did not propagate to the implementation (see Section 5.1 for more details). We provide the correct derivation in Eq. 6 as follows:

$$\text{tr}\{\Sigma_D^{-1}\Sigma_Q\} = \sum_{i=1}^{k} \frac{\sigma_{i_Q}^2}{\sigma_{i_D}^2}. \tag{6}$$

Finally, for the third term we have,

$$(\mu_Q - \mu_D)^\intercal \Sigma_D^{-1}(\mu_Q - \mu_D) = \sum_{i=1}^{k} \frac{(\mu_{i_Q} - \mu_{i_D})^2}{\sigma_{i_D}^2} = \sum_{i=1}^{k} \frac{\mu_{i_Q}^2}{\sigma_{i_D}^2} - \sum_{i=1}^{k} \frac{2\mu_{i_Q}\mu_{i_D}}{\sigma_{i_D}^2} + \sum_{i=1}^{k} \frac{\mu_{i_D}^2}{\sigma_{i_D}^2}. \tag{7}$$

Combining Eq. 5, 6 and 7 into Eq. 4, and removing constants, we arrive at the intended derivation of the ranking function:

$$\text{KLD}(Q\|D) = \sum_{i=1}^{k} \log \sigma_{i_D}^2 + \sum_{i=1}^{k} \frac{\sigma_{i_Q}^2}{\sigma_{i_D}^2} + \sum_{i=1}^{k} \frac{\mu_{i_Q}^2}{\sigma_{i_D}^2} - \sum_{i=1}^{k} \frac{2\mu_{i_Q}\mu_{i_D}}{\sigma_{i_D}^2} + \sum_{i=1}^{k} \frac{\mu_{i_D}^2}{\sigma_{i_D}^2}. \tag{8}$$

Note that, unlike Eq. 4, Eq. 8 is no longer the KL divergence. After all the simplifications, it is a KL divergence-based relevance scoring function for ranking documents, given a query. From this point forward, we continue with the work described in the original paper, but we base it on our Eq. 8, which is the derivation of the relevance scoring function that includes our correction.

The next step of this reproducibility study is to express Eq. 8 as a dot product between query and passage vectors,

$$\text{KLD}(Q\|D) = q^\intercal \cdot d, \tag{9}$$

with the purpose of reusing standard efficient inner product similarity search (Johnson et al., 2021). To do so, we isolate the document-specific terms of Eq. 8 that can be pre-computed:

$$\gamma_D = \sum_{i=1}^{k} \left( \log \sigma_{i_D}^2 + \frac{\mu_{i_D}^2}{\sigma_{i_D}^2} \right). \tag{10}$$

In the original paper, the term $\gamma_D$ is referred to as a "document prior". Now we can express the relevance score as a dot product between query and passage vector representations:

$$\vec{q} = \left[ 1, \sigma_{1_Q}^2, \ldots, \sigma_{k_Q}^2, \mu_{1_Q}^2, \ldots, \mu_{k_Q}^2, \mu_{1_Q}, \ldots, \mu_{k_Q} \right], \tag{11}$$

$$\vec{d} = \left[ \gamma_D, \frac{1}{\sigma_{1_D}^2}, \ldots, \frac{1}{\sigma_{k_D}^2}, \frac{1}{\sigma_{1_D}^2}, \ldots, \frac{1}{\sigma_{k_D}^2}, -\frac{2\mu_{1_D}}{\sigma_{1_D}^2}, \ldots, -\frac{2\mu_{k_D}}{\sigma_{k_D}^2} \right], \tag{12}$$

where $\vec{q}, \vec{d} \in \mathbb{R}^{1 \times (3k+1)}$. At this point we remind the reader that, following Eq. 3, the relevance score is the negative of Eq. 9.

## 3.2  Listwise knowledge distillation

Knowledge distillation has shown to be of great importance in boosting the effectiveness of dense retrievers. In detail, a highly effective cross-encoder re-ranker is used as a teacher to transfer knowledge to a less effective but efficient first-stage dense retriever student model. Consequently, the effectiveness of the dense retriever is increased while it retains its efficiency. That said, in the original work by Zamani & Bendersky (2023), the authors employ a listwise distillation loss function (Zeng et al., 2022) to train their dense retriever (i.e.,

student model). For each query $q$ and its set of passage $D_q$ (see Section 4.3 for details on how this set is constructed), the loss is computed as:

$$\sum_{d,d' \in D_q} \mathbb{1}\{y_q^t(d) > y_q^t(d')\} | \frac{1}{\pi_q(d)} - \frac{1}{\pi_q(d')} | \log(1 + e^{M_\theta(q,d') - M_\theta(q,d)}), \tag{13}$$

where $\pi_q(d)$ denotes the position of passage $d$ in the ranked list produced by the dense retrieval student model $M_\theta$ and $y_q^t(d)$ denotes the relevance judgment produced by the teacher model for the pair of query $q$ and passage $d$; $y_q^t(d)$ can be either a score or a label (see Section 4.3 for details).

## 4 Experimental Setup

### 4.1 Datasets and metrics

Our evaluation is performed on both in-domain (ID) data, and out-of-domain (OOD) data in a zero-shot setting. All tasks are ad-hoc retrieval, with a fixed set of documents. Statistics of the datasets are reported in Appendix C. We summarize the datasets and evaluation methodology below.

**In Domain (ID).** We train all models on the MS-MARCO Nguyen et al. (2016) training set. Note that we split the full training set into a training and validation set for hyperparameter tuning as described in Section 4.2. There are three in-domain evaluation sets, all of which are based on the MS-MARCO corpus. This includes the MS-MARCO Dev set, the TREC-DL 2019 Craswell et al. (2020) and TREC-DL 2020 Craswell et al. (2021) datasets. Both TREC datasets are densely labeled by humans. The evaluation metric for the Dev set is the mean reciprocal rank (MRR) with a cut-off of 10, denoted as MRR@10. For the TREC subsets, we use the standard evaluation metrics of normalized discounted cumulative gain at 10 (nDCG@10), and mean average precision (MAP).

**Out of Domain (OOD).** We evaluate the retrieval models' generalization ability in different domains via zero-shot passage retrieval experimentation. All retrieval models are trained on the MS-MARCO training set and tested on previously unseen queries and underlying corpus. We replicate the evaluation setup outlined in Zamani & Bendersky (2023), with nDCG@10 as the primary metric. We evaluate the following OOD datasets in zero-shot setting: (i) Scifact (Wadden et al., 2020): a scientific claim verification dataset where the task involves retrieving abstracts that either refute or support a claim, (ii) FiQA (Maia et al., 2018): a dataset that involves retrieval of documents in the financial domain using natural language questions, (iii) TREC-COVID (Voorhees et al., 2021): a biomedical dataset of scientific articles about COVID-19, with questions as the topics/queries, and (iv) CQADupStack (Hoogeveen et al., 2015): a community question answering (CQA) dataset, with the task of retrieving duplicate questions in a community website (StackOverflow).

### 4.2 Baselines

We compare MRL against the following single-vector dense retrieval models:

- **DPR** (Karpukhin et al., 2020): is a traditional dense retriever that is trained with softmax cross-entropy.

- **TAS-B** (Hofstätter et al., 2021): is an effective dense retriever that is trained by combining (i) knowledge distillation from a re-ranker teacher model (i.e., cross-encoder) with (ii) a balanced topic-aware sampling method. This method alternates the creation process of the training batches by composing batches based on queries clustered in the same topic. Furthermore, it selects passage w.r.t the pairwise margin between positive and negative passages in the batch so that the margin of positive-negative pairs is balanced in the margin range.

- **CLDRD** (Zeng et al., 2022): is a state-of-the-art dense retriever that uses TAS-B as initialization and is trained by combining curriculum learning with knowledge distillation; in particular, it uses the listwise loss of Eq. 13. The student dense retriever is trained via an iterative training process in which the difficulty of the training data, produced by the re-ranking teacher model, increases with each iteration.

The motivation behind selecting these baselines is twofold: First, their inclusion in the original study, and second, to enable fair comparisons in our subsequent ablation study. For instance, MRL can be compared with CLDRD to assess the impact of the multivariate representations, and a similar assessment can be made when MRL without distillation is compared against DPR.

### 4.3 Training setup

A crucial aspect of training MRL is the computation of the listwise distillation loss. In the original work of Zamani & Bendersky (2023), it is suggested that for computing the listwise distillation loss in Eq. 13:

- Given a query $q$, the passage set $D_q$ is constructed with positive passages provided by the dataset's official relevance judgments. On the other hand, the negative passages are sampled from the top-k passages retrieved with BM25 and the top-k passages retrieved by the student dense retrieval model itself (using an asynchronously updated ANN index).

- $y_q^t(d)$ is the raw score from the teacher model for a query-passage pair.

- Follow in-batch negative training to reuse passages from other queries that are already in the batch.

In addition to the training setup outlined above, we experiment with an alternative training perspective based on the intuition that knowledge distillation is primarily designed to leverage incomplete relevance judgements (rather than effective utilization of in-batch negatives) common in large-scale retrieval datasets like MS-MARCO Nguyen et al. (2016). That is, an effective teacher model might be preferred to using (random) in-batch negatives that might be easy to distinguish Hofstätter et al. (2021). Furthermore, in-batch negatives are expensive to compute for expensive models e.g., cross-encoders (Lin et al., 2021).

Exact relevance scores produced by the teacher do not impact the loss as it only considers the ordering rather than the score. Despite this, the scores control which query-passage pairs will contribute to the loss via the $\mathbb{1}\{y_q^t(d) > y_q^t(d')\}$ term. Therefore, when raw teacher scores are used, we argue that *all* pairs – even irrelevant or easy to distinguish pairs – contribute to the loss, which is a source of noise that pushes the student model to learn this possibly uninformative ordering. To combat this problem, Zeng et al. (2022) utilize labels instead of raw scores – all irrelevant passages are assigned the same label, removing comparisons between irrelevant pairs in the loss.

Motivated by the arguments outlined above, we follow the pseudo-labeling approach from CLDRD (Zeng et al., 2022). Therefore, to compute the listwise distillation loss:

- Given a query $q$, the passage set $D_q$ is constructed with respect to the top-k passages in the ranked list returned by the teacher model (reranking order). In particular, the first $K$ passages in the ranked list returned by the teacher model are considered positive, the next $K'$ are considered hard negatives, and the remaining $K''$ soft negatives.

- $y_q^t(d)$ is a relevance label according to the group, passage $d$ belongs to:

$$y_q^t(d) = \begin{cases} \frac{1}{r_{qd}^t} & \text{iff d is positive} \\ 0 & \text{iff d is hard-negative} \\ -1 & \text{iff d is soft-negative} \end{cases}$$

where $r_{qd}^t$ is the ranking position of the document $d$ given the query $q$ in the teacher ranked list. To this extent, we proceed with incorporating the curriculum formulation of CLDRD (Zeng et al., 2022) in MRL, thus following the exact training setup of CLDRD. The advantages of following this setup are two-fold: (i) preliminary results showed superior retrieval performance for the case where we employ the training scheme of CLDRD rather than the one suggested by the original MRL paper (see Section 5.3), and (ii) we create a fair comparison against CLDRD which is the primary competing approach. This approach allows us to attribute any observed performance increase solely to MRL's multivariate representations and not other

Table 1: Model specifications of MRL variants and CLDRD. Models are initialized with TAS-B and use the listwise knowledge distillation loss of Equation 13.

| | Representations | Passages | Teacher judgments | In-batch negatives | Curriculum Learning |
|---|---|---|---|---|---|
| CLDRD | Vector | Pseudo-labeling | Labels | No | Yes |
| MRL-Orig. | Distribution | positives: gold annotations negatives: BM25, ANN | Raw scores | Yes | No |
| MRL-Ours | Distribution | Pseudo-labeling | Labels | No | Yes |

aspects, such as differences in training data or teacher models. For the remainder of this work, we will refer to the MRL implementation that utilizes the original training setup as "MRL-Orig", and the implementation that utilizes the CLDRD training setup a "MRL-Ours". The differences between CLDRD, MRL-Ours and MRL-Orig are summarized in Table 1.

### 4.4 Query performance prediction

The QPP task (He & Ounis, 2004; 2006; Carmel & Yom-Tov, 2010) involves inferring the difficulty of a given query for a search system without using relevance judgments. We replicate the pre-retrieval QPP setup in Zamani & Bendersky (2023), evaluating on the TREC-DL 19 and TREC-DL 20 datasets. That is, we retrieve documents for given a query using a search system, and evaluate using nDCG@10 it to obtain a ground-truth assessment of performance. Then, we use a QPP method to predict the performance and evaluate it against the ground-truth assessment using three correlation measures, Spearman's correlation, Pearson correlation and Kendall's Tau.

The effectiveness of a QPP method is a function of the underlying retrieval system. Since it was unclear from the original study which system was used to compute the ground truth performance, we experiment with multiple search systems. In addition to the model itself, we experiment with three retrieval models independent of the MRL model, to measure how well the QPP method generalizes. We include a traditional lexical retriever (BM25), a simple dense retriever (DPR), and an effective dense retriever (TAS-B). We utilize the following baselines used in the original study:

- **SCQ** (Zhao et al., 2008): computes the similarity between a query and the corpus for each query term based on the frequency of occurrence of the term in the corpus.

- **VAR** (Carmel & Yom-Tov, 2010): considers the variance or standard-deviation of the term weights of each query term, based on the documents in which the term occurs.

- **IDF** (Carmel & Yom-Tov, 2010): is based on the inverse document frequency of each query term.

- **PMI** (Hauff, 2010): is a predictor that computes the pointwise mutual information, assigning high scores for frequently co-occurring query terms. Given all possible query term pairs, either the average or the maximum can be used as the predictor.

For SCQ, VAR, and IDF, the scores are computed at the query term level and then aggregated using either summing, averaging, or taking the maximum of each score. We report each of these aggregations in the results.

**QPP for MRL.** Zamani & Bendersky (2023) mention that the norm of the variance $|\Sigma_Q|$ is used to compute the predicted performance. We interpret this statement as using a *function* of the covariance, and use the negative norm $-|\Sigma_Q|$, because the predicted variance should *increase* for queries that are difficult, rather than decrease. For instance, a typographical error in the query makes it more difficult to address than a "clean" query (Sidiropoulos & Kanoulas, 2022; 2024), leading to lower performance compared to a clean query. Similarly, an OOD query could also result in poorer performance on average compared to an ID query. From the QPP perspective, a model should therefore assign *lower* predicted performance for these types of queries, motivating our choice. In Section 5.2.1, we show that this intuition holds empirically.

### 4.5 Implementation details

As mentioned above, we follow the exact training setup as in CLDRD. This setup includes three curriculum learning iterations with 100K, 50K, and 50K steps for the first, second, and third iterations. The number of passages for each query is 30. We use 5 positives, 12 hard negatives, and 13 soft negatives for the first iteration. Similarly, for the second iteration, we use 10, 10, and 10 positives, hard-negatives, and soft-negatives, respectively, while the third iteration consists of 30 positives only. We have a batch size of 15, the maximum we can fit in a 40GB A100 GPU. We set the maximum length for queries and passages to 32 and 256 tokens, respectively. We initialize the dense retriever student model with the official TAS-B checkpoint, and we set as the teacher model the `ms-marco-MiniLM-L-6-v2` cross-encoder that is publicly available on HuggingFace. The learning rate is set to $[5 \times 10^{-6}, 1 \times 10^{-6}, 1 \times 10^{-6}]$ for the three iterations using Adam, and the rate of the linear scheduling with a warm-up is set to 0.1. The $\beta$ parameter for softplus is set to 2.5. For MRL, the mean and variance are obtained by passing the CLS token and a VAR token respectively through fully connected projection layers. The MRL models reported use means and variances projected down to 383 $(= \frac{768}{2} - 1)$.

Since MS-MARCO does not come with a validation set, we split the train set into a validation (6890 queries) and train set. The parameters above were selected after a hyperparameter search with the validation set performance used to pick the best model. Refer to Appendix D for the full set of hyperparameters. We use the `Tevatron` toolkit (Gao et al., 2023) to train the models and the `pytrec_eval` library (Van Gysel & de Rijke, 2018) to evaluate the retrieval performance. Finally, we our QPP baselines are based on an existing implementation by Meng et al. (2023).

## 5 Discussion

We organize the discussion section around retrieval experiments in Section 5.1, the investigation of the variance vectors in Section 5.2, and the results of the ablation study in Section 5.3.

### 5.1 Reproducing the retrieval results

We start by testing whether we can obtain the results reported in the original study. We report the results in Table 2, where DPR, CLDRD, and MRL are our implementations of the original methods. We report results for TAS-B by utilizing the official pre-trained checkpoint, which also serves as CLDRD and MRL initialization. This way, we can ensure a fair comparison between the different methods. We include the original numbers in the lower group in Table 2. At this point, we want to underline that for MRL we use the corrected KL formulation we presented in Section 3 for our experiments. Our decision to do so is grounded in the belief that the original study's authors also utilized this formulation in their implementation and that the formulation with the mathematical error is a typographical mistake in their paper. We arrived at this conclusion based on our preliminary experiments, which yielded a dramatically low retrieval performance (i.e., MRR@10 was 0.134 for MS-MARCO) when following the wrong formulation.

We first focus on testing whether we are able to replicate the results for CLDRD, the main counterpart of MRL. As shown in Table 2, our experimental results affirm the state-of-the-art retrieval performance (for single-vector dense retrievers) of CLDRD. Furthermore, our findings validate the original study regarding its ability to enhance the performance of TAS-B, both in the ID and OOD scenarios. We consider this a successful replication despite the slight discrepancy in the results. The reason for not obtaining the exact same results can be attributed to different development toolkits, hardware, or implementation details not present in the original work, etc.

Regarding the reproduction of MRL, as shown in Table 2, we were unable to yield the same results as the original study. First and foremost, we notice that following the original training setup for MRL cannot train an effective retriever (see MRL-Orig. in Table 2). Specifically, there is a discrepancy of 0.138 in MRR@10 between our replicated model and the original reported values. In Table 2, we showcase that MRL-Ours, which adopts an alternative training approach (outlined in Section 4.3), can successfully reduce the gap in performance against the reported MRL to 0.018.

Table 2: Reproduction results of MRL. The upper part contains the reproduction results, while the lower part contains the results reported in the original study. MRL-Ours indicates that our proposed training setup for MRL is being followed, while MRL-Orig. denotes adherence to the training setup outlined in the original paper. In a fair comparison, MRL fails to outperform the main competing approach, CLDRD.

| | Model | MS MARCO | | TREC-DL'19 | | TREC-DL'20 | | SciFact | FiQA | TREC-COVID | CQADupStack |
|---|---|---|---|---|---|---|---|---|---|---|---|
| | | MRR@10 | MAP | NDCG@10 | MAP | NDCG@10 | MAP | NDCG@10 | NDCG@10 | NDCG@10 | NDCG@10 |
| Reproduced | DPR | .312 | .319 | .649 | .345 | .625 | .356 | .474 | .231 | .600 | .266 |
| | TAS-B | .344 | .351 | .721 | .396 | **.685** | .430 | **.643** | .301 | .481 | .313 |
| | CLDRD | **.378** | **.383** | **.727** | **.448** | .670 | **.446** | .627 | **.308** | **.608** | **.327** |
| | MRL-Orig. | .255 | .261 | .576 | .270 | .534 | .304 | .305 | .146 | .169 | .185 |
| | MRL-Ours | .375 | .380 | .721 | .439 | .667 | .438 | .605 | .293 | .510 | .320 |
| Reported | TAS-B | .344 | .351 | .717 | .447 | .685 | .455 | .643 | .300 | .481 | .314 |
| | CLDRD | .382 | .386 | .725 | .453 | .687 | .465 | .637 | .348 | .571 | .327 |
| | MRL | .393 | .402 | .738 | .472 | .701 | .479 | .683 | .371 | .668 | .341 |

Even with our best-performing MRL, namely, MRL-Ours, we observe a drop in performance across all reported metrics for both the ID and OOD datasets. For instance, in the case of FiQA, our implementation yielded an NDCG@10 of 0.293, which is significantly lower than the original study's reported value of 0.371. Furthermore, we could not find the trends that were reported in the original study. Specifically, the original study showed that MRL outperformed both TAS-B and CLDRD in ID and OOD scenarios. In contrast, in our work, MRL achieves similar performance to CLDRD in the ID datasets. A similar trend holds for the OOD dataset, except for TREC-COVID, where CLDRD outperforms MRL with a substantially higher score of 0.608 compared to 0.510 for MRL. Upon comparing MRL with TAS-B, it is noted that MRL either matches or outperforms TAS-B in the ID datasets, except for TREC-DL 20. However, in the OOD datasets, MRL surpasses TAS-B only for TREC-COVID and CQADupStack.

We hypothesize that the original trends were not reproduced perhaps due to the different training/experimental setups for the baseline and the MRL model in the original study. For instance, CLDRD is trained with a batch size of 8, and the original study reports models trained with a batch size of 512.

Testing this hypothesis would require replicating the same training and experimental settings used in the original study. However, an exact replication of the initial work is not possible since the original manuscript does not sufficient details for the best hyper-parameters (e.g., learning rate, softplus $\beta$, number of training steps), the cross-encoder model that was used as a teacher, as well as the number of soft and hard negative passages per query in a batch. Additionally, it reports a rather large batch of 512, which is unattainable within the limitations of an academic GPU budget.

We stress that our goal is to reproduce the paper (i.e., generalize to a different training setup) rather than replicate, by using a training and experimental setup that facilitates a fair comparison of MRL against baseline models. As such, when comparing our implementation of MRL against CLDRD, performance differences can be attributed solely to MRL's approach of representing queries and passages as distributions. From our experimental results, we can conclude that although MRL is a competitive approach, the multivariate representations do not boost the retrieval performance. Furthermore, we unveil that MRL cannot consistently outperform its counterparts when evaluated under fair comparisons. Different from CLDRD, MRL produces a variance that can be utilized in downstream tasks. We investigate this predicted variance in the following section.

## 5.2 Analyzing the variance

We analyze the predicted variance in three experiments: query performance prediction experiments (Section 5.2.1), experiments with typos, and retrieval experiments with alternate encoding schemes in Section 5.2.2. The first is a replication of the QPP experiments outlined in the original paper, while the latter two are additional analysis. We perform this analysis on MRL-Ours, our best-performing MRL model.

Table 3: QPP results for MRL for four different reference models. While MRL does perform well for TREC-DL 20 for BM25, it fails to do so for TREC-DL 19. The opposite is true for DPR. For both TAS-B and MRL reference models, MRL is more consistent but fails to reach the reported performance (bottom row). Furthermore, as is evident from Figures 1 and 2, MRL is outperformed by simple baselines.

|  |  | TREC-DL 19 | | | TREC-DL 20 | | |
|  |  | S-$\rho$ | P-$\rho$ | K-$\tau$ | S-$\rho$ | P-$\rho$ | K-$\tau$ |
|---|---|---|---|---|---|---|---|
| MRL | BM25 | .024 | .030 | .032 | .308 | .313 | .203 |
| | DPR | .222 | .190 | .158 | -.075 | -.011 | -.045 |
| | TAS-B | .124 | .190 | .075 | .139 | .155 | .093 |
| | MRL | .105 | .170 | .068 | .170 | .200 | .113 |
| | MRL (reported) | - | .271 | .259 | - | .272 | .298 |

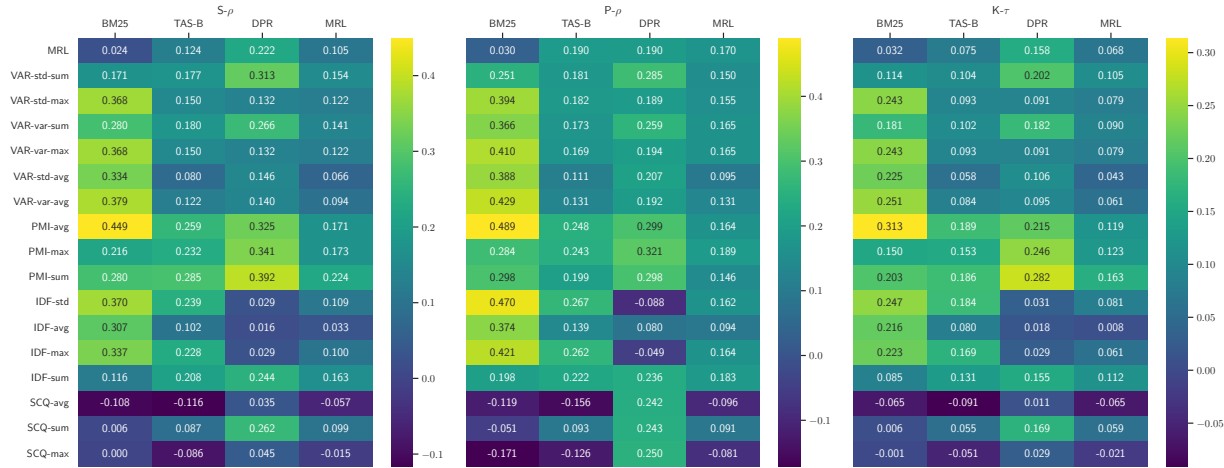

Figure 1: QPP results for TREC-DL 19 for four reference models (x-axis) and several methods (y-axis). Each subplot corresponds to a different correlation metric: Spearman's (S-$\rho$), Pearson's (P-$\rho$), and Kendall's Tau (K-$\tau$) correlations. MRL is outperformed by simple baselines regardless of the reference model or metric.

### 5.2.1 Query performance prediction

The results for the QPP experiments are plotted in Figures 1 and 2. We also report the MRL results separately in Table 3, which also includes the reported numbers in Zamani & Bendersky (2023). As mentioned previously, we utilized four different reference models because the original study did not report which model was used, and also to see if MRL generalizes to different reference models. We note that the original paper does not mention how the norm is used in computing the predicted performance – in our experiments we use the *negative* norm (using the norm flips the signs of the correlation) – intuitively, a higher uncertainty should result in lower performance (see Section 4.4).

From Table 3, we were unable to reproduce the numbers for MRL reported in the original paper, with any of the reference models. While we do achieve higher than reported correlations for TREC-DL 20 with BM25, we note that MRL fails to generalize to TREC-DL 19. This observation is flipped for DPR, where MRL performs well for TREC-DL 19 but not TREC-DL 20. For TA-B and DPR, we see that MRL is more consistent. But how does MRL compare with the baselines?

Figure 1 contains results for TREC-DL 19, with each subplot corresponding to the three metrics we used. Comparing the MRL (top row) with the other methods, we notice that a simple baseline outperforms MRL for each metric regardless of the reference metric. In particular, at least one variant of the PMI baseline

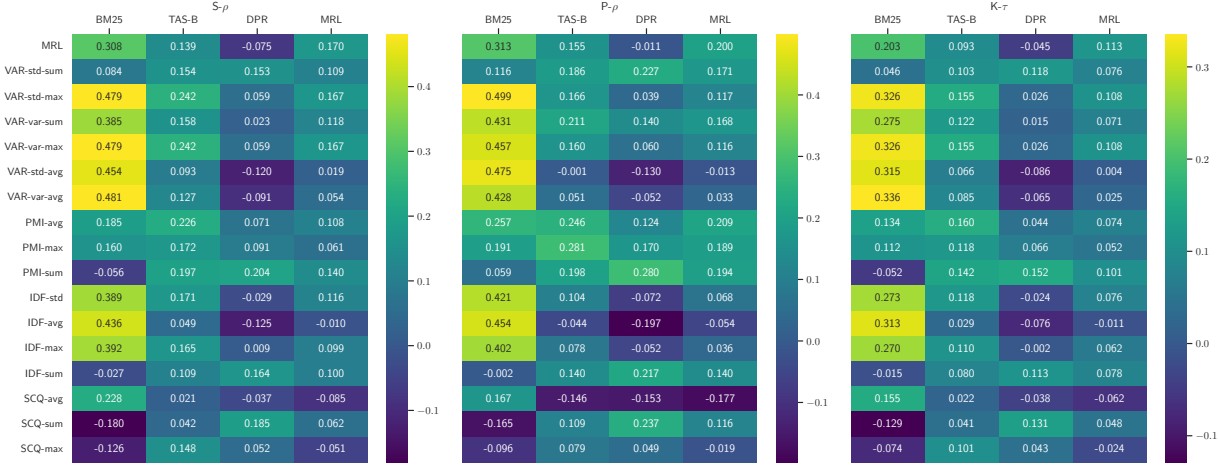

Figure 2: QPP results for TREC-DL 20. Simple lexical baselines outperform MRL when the reference is BM25, TAS-B, or DPR. With MRL itself as the reference, MRL achieves similar or better performance.

outperforms MRL. We remind the reader that these baselines are simple, non-parametric methods that use statistics derived from the corpora to compute the query difficulty.

For TREC-DL 20, the results are more encouraging. We observe similar trends for three reference models BM25, TAS-B and DPR, with MRL outperformed by at least one baseline (PMI, VAR, IDF). However, if the reference model is MRL itself, we observe that MRL achieves similar performance, or even beats every other baseline for one metric (K-$\tau$).

In these experiments, we investigate the degree to which the MRL framework captures the notion of uncertainty by measuring a proxy – query difficulty. Intuitively, this suggests that for some datasets and reference models, higher uncertainty was indeed assigned to difficult queries. However, the positive correlation is *not consistent*. MRL fails to generalize to different reference models and datasets, achieving random correlation in many settings.

Furthermore, even the positive correlations are typically weak, and MRL is outperformed by simple non-parametric baselines in most comparisons. The lack of consistent and strong correlations suggests that MRL is unlikely to be a strong and consistent predictor of query difficulty in our experimental setup. Motivated by these results, we explore what the predicted variance captures in the next section.

### 5.2.2 Does MRL capture uncertainty?

We outline two additional experiments investigating the predicted variance beyond the original paper: (a) contrasting the predicted variance of corrupted and clean data and (b) experimenting with alternate encoding schemes.

**Experiments with typographical errors.** Here, we consider an analog to the QPP experiments above, but instead of retrieval difficulty, we examine if the model is sensitive to *data distribution shifts* instead. Inspired by several works in the vision domain (e.g., Warburg et al., 2021; 2023), we argue that a model should assign higher variance to *corrupted* or *OOD data* compared to clean or ID data.

We experiment with the DL-Typo (Zhuang & Zuccon, 2022) dataset that contains 60 query pairs accompanied by their relevance assessments. Each pair consists of a real user query with typographical errors and its corresponding version where these errors have been corrected. For instance, the corrupted query "what is acid reflex" and its typo-free version "what is acid reflux". Given these data, we compute the norms of the predicted variance and plot their distributions. If the model does indeed model uncertainty accurately, we expect (a) clean data should be assigned lower variance compared to corrupted data (b) the distributions

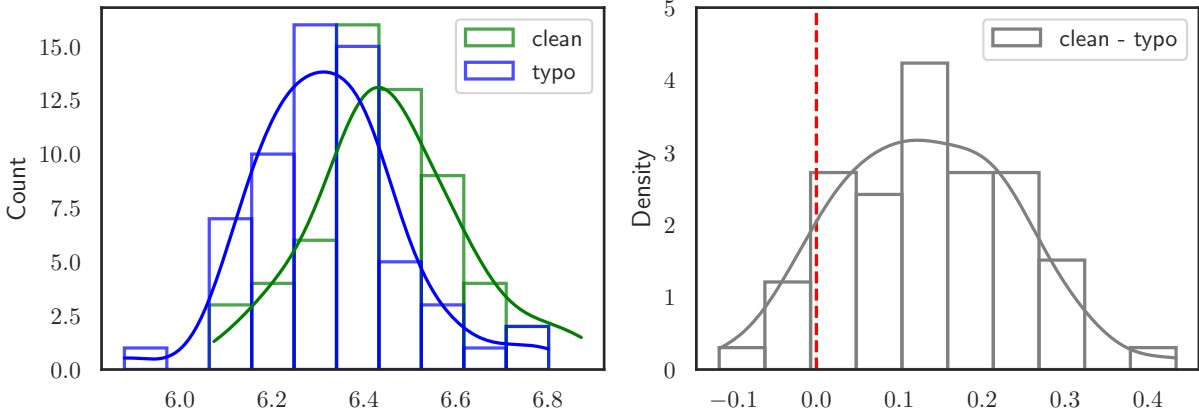

Figure 3: Uncertainty of clean and corrupted data: We plot the distributions of the norm of the predicted variance of the clean and corrupted data on the left. MRL assigns *lower* uncertainty to the corrupted data compared to the clean data. On the right, we plot the distribution of $|\Sigma_{clean}| - |\Sigma_{corrupted}|$, which is positive. Contrary to expectations, MRL fails to assign higher uncertainty to corrupted data.

Table 4: Performance of different encoding schemes for the MRL model. The first row uses the original equations proposed in the original paper, whereas the second row includes our corrections.

| Encoding | MS MARCO | | TREC-DL'19 | | TREC-DL'20 | | SciFact | FiQA | TREC-COVID | CQADupStack |
|---|---|---|---|---|---|---|---|---|---|---|
| | MRR@10 | MAP | NDCG@10 | MAP | NDCG@10 | MAP | NDCG@10 | NDCG@10 | NDCG@10 | NDCG@10 |
| Zamani & Bendersky (2023) (Eq. 16 & 17) | .134 | .140 | .347 | .109 | .340 | .134 | .014 | .056 | .132 | .099 |
| Ours (Eq. 11 & 12) | .375 | .380 | .721 | .439 | .667 | .438 | .605 | .293 | .510 | .320 |
| Mean | .242 | .248 | .562 | .251 | .493 | .233 | .291 | .142 | .115 | .174 |

of the clean and corrupted data are well separated (c) the differences between the clean and corrupted data i.e., $|\Sigma_{clean}| - |\Sigma_{corrupted}|$, should be negative.

We plot these distributions in Figure 3. We note while we expected more variance to be assigned to the corrupted data, the opposite is true. While there is some separation observed between the two distributions – the distribution of the corrupted data is to the left of the clean data. The right plot underscores this result, as the $|\Sigma_{clean}| - |\Sigma_{corrupted}|$ distribution is mostly positive. This suggests that contrary to expectation, the model predicts a higher variance for clean data. We expand on this analysis by examining the role of the predicted variance in retrieval relevance.

**Encoding.** MRL produces both a mean and variance for queries and documents. In Table 2, we reported retrieval results using the encoding scheme which enables retrieval using the KL divergence i.e., documents are encoded and indexed with Equation 12 and queries with Equation 11. If the variance only captures uncertainty, we argue that the difference in retrieval performance when only the mean is used should not be much lower than this encoding scheme. However, as we show in Table 4, this is not true. Comparing the encoding scheme (second row) with using just the mean (third row), we see that performance drops sharply across all datasets.

The performance drop could be explained partly due to the way the models were trained. Since the full KL loss (Equation 4) was used in training the model, it may not be equipped to perform retrieval using just the mean. However, intuitively, we expect the mean to model *relevance* and variance to model *uncertainty*, which means that the retrieval performance should not be drastically different when only the mean is utilized for retrieval. The drastic drop suggests that the model may be instead utilizing the predicted variance vectors as a signal for relevance. This is the core difference between a method such as MRL and a Bayesian method:

Table 5: Ablation study on the different sources of performance improvements in the MRL framework. The first row indicates MRL following our training setup, while the last row is MRL with the original setup. Performance is greatly influenced by the initialization (TAS-B), the loss (Listwise KD), as well as how the training batch is constructed.

| | MS MARCO MRR@10 | TREC-DL'19 NDCG@10 | TREC-DL'20 NDCG@10 | SciFact NDCG@10 | FiQA NDCG@10 | TREC-COVID NDCG@10 | CQADupStack NDCG@10 |
|---|---|---|---|---|---|---|---|
| MRL-Ours
 Multivariate representation
 TAS-B
 Listwise KD
 Teacher constructs the batch
 Teacher pseudolabels | .375 | .721 | .667 | .605 | .293 | .510 | .320 |
| - Multivariate representation
+ Vector representation
(i.e., CLDRD) | .378 | .727 | .670 | .627 | .308 | .608 | .327 |
| - TAS-B
+ DistilBERT | .349 | .684 | .650 | .539 | .263 | .551 | .307 |
| - Teacher constructs the batch
+ Qrels & negative mining | .331 | .681 | .659 | .461 | .224 | .402 | .272 |
| - Listwise KD
+ Cross-entropy | .328 | .629 | .644 | .498 | .245 | .473 | .268 |
| - TAS-B
- Listwise KD
- Multivariate representation
+ DistilBERT
+ Cross-entropy
+ Vector representation | .315 | .636 | .631 | .494 | .244 | .574 | .280 |
| - Teacher constructs the batch
- Teacher pseudolabels
+ Teacher raw scores
+ Qrels & negative mining
(i.e., MRL-Orig.) | .255 | .576 | .534 | .305 | .146 | .169 | .185 |

variance in MRL is not statistical variance, i.e., it does not express deviation from the mean prediction. Instead, variance in MRL is a deterministically estimated quantity that minimizes a distance objective.

In this section, we examined if the MRL model consistently predicts a variance that reflects a notion of uncertainty defined by either query difficulty or sensitivity to data distribution shifts. We find that the QPP results are inconsistent, and the model against initial expectations assigns a higher uncertainty to corrupted data. In addition, experiments with encoding using only the mean suggest that the variance seems to model relevance since performance drops drastically when only the mean is used for retrieval.

### 5.3 Ablation study

Even though the MRL performance reported in the original paper was not reproduced under our experimental setup, MRL remains a framework that can produce a highly effective dense retriever. MRL consists of several components: multivariate representations, knowledge distillation, model initialization from a pre-trained dense retriever, and a batch construction strategy. Thus, it needs to be made clear how much each component impacts effectiveness. To understand how each component affects the retrieval results, we experiment with several MRL variants. The ablations are performed on the MRL-Ours model, the best-performing MRL we could obtain and results are reported in Table 5.

**Multivariate representations.** MRL represents queries and passages as multivariate distributions and uses negative multivariate KL divergence to compute similarity. First and foremost, we want to understand how much the multivariate representations contribute to the overall effectiveness of the retriever. In order to test this, we conduct an experiment where we substitute multivariate representations with vector representations and compute similarity via the dot product. We find that multivariate representations do not lead to a higher retrieval performance; in contrast, we observe a small decrease in performance when utilized.

**Model initialization.** An important question regarding MRL's retrieval performance is how much it will degrade if not initialized with a pre-trained TAS-B checkpoint. To test this, we use DistilBERT as the

initialization instead. When comparing the first two rows in Table 5, we see that it is possible to train a competitive model with a DistilBERT initialization; however, it cannot achieve state-of-the-art performance.

**Training batch construction.** We explore different approaches for constructing the training batch, namely, using the teacher model to produce the data vs. obtaining it through the ground-truth query relevance judgments (i.e., qrel file) and negative mining; positive passages come from human judgments, while negative passages are retrieved from BM25 and an asynchronous ANN index (see Section 4.3). The former is often used alongside knowledge distillation training since it can overcome the sparsity issue of the ground-truth query relevance judgments files by relying only on the teacher's signal. The latter is commonly used in combination with a softmax cross-entropy loss. Our results in Table 5 (comparing the first with the third row) indicate that relying on human judgments and negative mining for constructing the training data decreases performance significantly.

**Training strategy.** Besides training with knowledge distillation, MRL can be trained with supervised contrastive learning by minimizing a softmax cross-entropy loss. We test this alternative and find that following a knowledge distillation training scheme is crucial for training an effective retriever. This result is in line with recent works in dense retrieval that have shown the superiority of knowledge distillation training over supervised contrastive learning (Hofstätter et al., 2021; Lin et al., 2021).

**MRL-Ours vs. MRL-Orig.** When training with knowledge distillation, the teacher's relevance judgments can be either pseudo-labels or raw scores (controlling term $\mathbb{1}\{y_q^t(d) > y_q^t(d')\}$ in Equation 13). Focusing on the first, fourth, and last rows in Table 5, we can see the dramatic effect of using raw scores. This low performance is likely related to the fact that the loss term, which controls pairs will contribute to the loss, is bypassed when raw scores are utilized. As a result, all possible pairs, including those of irrelevant passages, contribute to the loss computation, forcing the student model to match the exact ordering the teacher model provides between irrelevant passages (see Section 4.3).

At this point, we want to underline that MRL-Ours consists of training with a listwise knowledge distillation loss with pseudo-labels for teacher judgments, using the cross-encoder teacher to construct the batches, and TAS-B initialization. In contrast, MRL-Orig. follows the original MRL paper that suggests constructing the batches via the ground-truth query relevance judgments and negative mining (BM25+ANN) and using raw scores as the judgments from the teacher model. Our experiments have revealed that adhering to the MRL-Orig training in our experimental setup leads to dramatically low retrieval performance. We empirically demonstrate that relying on ground-truth query relevance judgments and negative mining for batch construction, as well as depending solely on raw teacher scores, performs worse than their alternatives.

## 5.4 Simple extension to reduce the hyperparameter search space

The MRL model produces a mean and a diagonal co-variance matrix / variance vector given text input. Using the raw co-variance without ensuring that it is positive (and semi-definite) would make the model produce invalid values for the variance e.g., a negative variance. To ensure positivity, Zamani & Bendersky (2023) pass the raw values through a *softplus* activation function ($\text{Softplus}(x) = \frac{1}{\beta} * \log(1 + \exp(\beta * x))$), which ensures that the predicted variance is positive. The $\beta$ parameter in the *softplus* function as a hyperparameter, with the original study notes that the retrieval performance is robust to $\beta$. An alternative to using the *softplus* function that predicts a variance is to predict the *log*-variance instead, without any activation function. That is, we assume that the raw values produced by the encoder is the log-variance. As such, these values are exponentiated prior to use e.g., when plugging it into Equations 12 and 11. This alternate method is simpler, and renders the $\beta$ hyperparameter unnecessary. We term this variant "LogVar", and validate this approach through experimentation.

Our experiments include the MRL model trained with and without distillation, which is compared to the *softplus* variant. We stress that we do not expect improved performance; this is a variant that functions similar to the original model. The results, reported in Table 6, show that the LogVar models achieve similar performance across datasets and metrics. We also conducted an ablation with different initializations and observe the same trend. In essence, the LogVar model provides similar results to the Softplus model without requiring an additional hyperparameter.

Table 6: MRL Variant: LogVar vs Softplus: LogVar performs similarly to Softplus, both for the MRL model and MRL without distillation (denoted as MRL (-LKD +CE)), but with no hyperparameter tuning required.

| | Model | MS MARCO MRR@10 | TREC-DL'19 NDCG@10 | TREC-DL'20 NDCG@10 | SciFact NDCG@10 | FiQA NDCG@10 | TREC-COVID NDCG@10 | CQADupStack NDCG@10 |
|---|---|---|---|---|---|---|---|---|
| Softplus | MRL | .375 | .721 | .667 | .605 | .293 | .510 | .320 |
| | MRL (-LKD +CE) | .328 | .629 | .644 | .498 | .245 | .473 | .268 |
| LogVar | MRL | .372 | .714 | .673 | .610 | .299 | .510 | .321 |
| | MRL (-LKD +CE) | .330 | .641 | .647 | .523 | .241 | .342 | .265 |

## 5.5 Limitations of our study

While we attempted a faithful reproducibility study, there were a handful of factors that limited the scope of our work. In this section we outline these limitations and our best efforts to mitigate them.

**Lack of details.** As mentioned previously, our reproducibility effort was hampered by a lack of crucial details in the original paper, such as the composition of the batches used while training the model, or which underlying cross-encoder was used as a teacher. Therefore, it is unclear whether the impressive performance improvements reported in the original paper are the result of some crucial implementation detail that was omitted. However, we tried to mitigate this limitation through *exhaustive* experimentation. The most promising part of that experimentation is presented in our ablation study, but it still constitutes a fraction of all model configurations that we attempted in order to get closer to the reported performance.

**Hyperparameter search.** Our access to realistic hardware resources allowed us to perform hyperparameter search only on the models in Table 2 and the model using cross-entropy without distillation in Table 5. We outline these hyperparameters in Appendix D. However, we note that the model performance is quite robust to the choice of hyperparameters in our initial experiments – mitigating this limitation to an extent.

**Batch size.** As mentioned in Section 4.5, we limited the batch size to 15, which was the maximum batch size possible on one GPU. This is in contrast to the original batch size of 512. However, we note that (a) experiments with more GPUs led to similar results (b) limiting to a batch size of 15 across all models makes it a fair comparison, especially given the impact of batch size on downstream retrieval performance (Sidiropoulos et al., 2021; Gao et al., 2021).

## 6 Conclusion

In this paper, we reproduce the multivariate representation learning framework by Zamani & Bendersky (2023). After addressing a likely typographical error in the original paper's derivations, we show that in a fair comparison, MRL fails to outperform baseline models during retrieval, both for in- and out-of-domain datasets. While MRL does not outperform baselines, we maintain that it still remains a competitive retrieval model. We also conduct an extensive analysis of the predicted variance. Against our expectations, our analysis reveals that the variance vectors do not consistently express uncertainty. In contrast to the original paper, we conduct a thorough ablation study, investigating the impact of the different components of the MRL framework: (i) multivariate representations, (ii) distillation, and (iii) model initialization. Through this study, we conclude that while multivariate representations do not harm performance, distillation is likely the primary source of improvement.

While we are unable to reproduce the results, we maintain that the ideas in the original paper are very valuable to the community. Prior to this paper, uncertainty was only utilized in ranking, not first-stage retrieval. The decomposition of the KL divergence as a dot-product enables incorporating uncertainty in first stage retrieval. This implies that *any* model that produces a distribution for queries/passages can be used in this framework, even if it was not trained with the objective function outlined in the paper – this is a promising direction of future research. Future work could further consider incorporating *document* uncertainty to the framework, e.g., for post-retrieval QPP.

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

## A Dot product formulation in the original paper

We include the original dot product formulation below of the KL divergence for completeness. Consider the term $\text{tr}\{\Sigma_D^{-1}\Sigma_Q\}$, the source of the error. Zamani & Bendersky (2023) formulate this as:

$$\text{tr}\{\Sigma_D^{-1}\Sigma_Q\} = \frac{\prod_{i=1}^{k}\sigma_{i_Q}^2}{\prod_{i=1}^{k}\sigma_{i_D}^2}. \tag{14}$$

This differs from our version in Equation 6. This error is propagated throughout the next steps. The KL formulation becomes:

$$\text{KLD}(Q\|D) = \sum_{i=1}^{k}\log\sigma_{i_D}^2 + \frac{\prod_{i=1}^{k}\sigma_{i_Q}^2}{\prod_{i=1}^{k}\sigma_{i_D}^2} + \sum_{i=1}^{k}\frac{\mu_{i_Q}^2}{\sigma_{i_D}^2} - \sum_{i=1}^{k}\frac{2\mu_{i_Q}\mu_{i_D}}{\sigma_{i_D}^2} + \sum_{i=1}^{k}\frac{\mu_{i_D}^2}{\sigma_{i_D}^2} \tag{15}$$

The equivalent dot product formulation, after taking the negative of the equation above gives us the original formulation (Equation (14) in Zamani & Bendersky (2023), with signs flipped for $\vec{d}$):

$$\vec{q} = \left[1, \prod_{i=1}^{k}\sigma_{i_Q}^2, \mu_{1_Q}^2, \ldots, \mu_{k_Q}^2, \mu_{1_Q}, \ldots, \mu_{k_Q}\right], \tag{16}$$

$$\vec{d} = \left[\gamma_D, \frac{1}{\prod_{i=1}^{k}\sigma_{i_D}^2}, \frac{1}{\sigma_{1_D}^2}, \ldots, \frac{1}{\sigma_{k_D}^2}, \frac{2\mu_{1_D}}{\sigma_{1_D}^2}, \ldots, \frac{2\mu_{k_D}}{\sigma_{k_D}^2}\right], \tag{17}$$

where $\gamma_D$ is equivalent in our formulation i.e., Equation 10. Here, $\vec{q}, \vec{d} \in \mathbb{R}^{1\times(2k+1)}$ and $q^\intercal \cdot d$ is equal to Equation 15. Note that the final vector for $\vec{d}$, the signs are flipped because we want to obtain the negative KL.

## B Dot product formulation of full KL divergence

We start from the *unsimplified* version of Eq. 8, that includes constants:

$$\text{KLD}(Q\|D) = \frac{1}{2}\left[\sum_{i=1}^{k}\log\sigma_{i_D}^2 - \sum_{i=1}^{k}\log\sigma_{i_Q}^2 - k + \sum_{i=1}^{k}\frac{\sigma_{i_Q}^2}{\sigma_{i_D}^2} + \sum_{i=1}^{k}\frac{\mu_{i_Q}^2}{\sigma_{i_D}^2} - \sum_{i=1}^{k}\frac{2\mu_{i_Q}\mu_{i_D}}{\sigma_{i_D}^2} + \sum_{i=1}^{k}\frac{\mu_{i_D}^2}{\sigma_{i_D}^2}\right]. \tag{18}$$

To formulate it as a dot product, we use the definition of the document prior, $\gamma_D$, from Eq. 10, but further define $\gamma_Q$ as:

$$\gamma_Q = \sum_{i=1}^{k}\log\sigma_{i_Q}^2. \tag{19}$$

Then we can extend the vector representations in Eq. 11 and Eq. 12 to include $\gamma_Q$:

$$\vec{q'} = \left[1, \gamma_Q, \sigma_{1_Q}^2, \ldots, \sigma_{k_Q}^2, \mu_{1_Q}^2, \ldots, \mu_{k_Q}^2, \mu_{1_Q}, \ldots, \mu_{k_Q}\right], \tag{20}$$

$$\vec{d'} = \left[\gamma_D, 1, \frac{1}{\sigma_{1_D}^2}, \ldots, \frac{1}{\sigma_{k_D}^2}, \frac{1}{\sigma_{1_D}^2}, \ldots, \frac{1}{\sigma_{k_D}^2}, -\frac{2\mu_{1_D}}{\sigma_{1_D}^2}, \ldots, -\frac{2\mu_{k_D}}{\sigma_{k_D}^2}\right], \tag{21}$$

where $\vec{q'}, \vec{d'} \in \mathbb{R}^{1\times(3k+2)}$. Then,

$$\text{KLD}(Q\|D) = \frac{1}{2}\left(\vec{q'}^\intercal \cdot \vec{d'} - k\right), \tag{22}$$

should precisely yield the KL divergence between the distributions of $Q$ and $D$, as defined in Eq. 4.

Table 7: Statistics and Description of Evaluation Datasets. Number of tokens for average query/document lengths were computed based on the `distilbert-base-uncased` tokenizer.

| | Name | Domain | # q | # p | avg. q length | avg. p length |
|---|---|---|---|---|---|---|
| ID | MS MARCO Dev | Miscellaneous | 6,890 | 8,841,823 | 9.01 | 76.97 |
| | TREC-DL 19 | Miscellaneous | 43 | 8,841,823 | 9.02 | 76.97 |
| | TREC-DL 20 | Miscellaneous | 54 | 8,841,823 | 9.22 | 76.97 |
| OOD | Scifact | Scientific Document Retrieval | 300 | 5,183 | 22.84 | 315.65 |
| | FiQA | Financial QA | 648 | 57,638 | 15.59 | 177.11 |
| | TREC-COVID | Biomedical document retrieval | 50 | 171,332 | 18.04 | 224.78 |
| | CQADupStack | Community QA retrieval | 13,145 | 457,199 | 13.55 | 248.73 |

Table 8: Hyperparameter search space for DPR (LR), CLDRD (LR) and MRL (LR, $\beta$). The best parameters were: MRL($\beta = 2.5, LR = [5 \times 10^{-6}, 1 \times 10^{-6}, 1 \times 10^{-6}]$), CLDRD ($LR = [7 \times 10^{-6}, 3 \times 10^{-6}, 3 \times 10^{-6}]$) and DPR ($LR = 7 \times 10^{-6}$).

| Parameter | Values |
|---|---|
| Learning Rate (LR) | $1 \times 10^{-4}, 1 \times 10^{-5}, 1 \times 10^{-6},$ $3 \times 10^{-6}, 5 \times 10^{-6}, 7 \times 10^{-6}$ |
| $\beta$ | 0.5, 1, 2.5, 7.5 |

## C  Datasets

In Table 7 we present the statistics of the datasets. Here, "p" and "q" indicate questions and passages, respectively. The length is in tokens.

## D  Hyperparameters

In Table 8 we detail our hyperparameter search space used throughout our experimental setup.

