# OpenReview forum: "Multivariate Dense Retrieval: A Reproducibility Study"
_TMLR — Rejected by TMLR_

### Review · Reviewer_mopx · 2024-05-16

**Summary Of Contributions:**

In this paper, the authors reproduce the MRL (Multivariate Representation Learning) framework for dense retrieval, and find some problems in the original work. They correct a mathematical error, and reproduce the code under the circumstance that no official code or checkpoint is released. They improve the training setup and provide a thorough ablation study. Through analysis and empirical results, the authors suggest that MRL actually failed to capture the uncertainty, and therefore the performance improvement stems from the model initialization and knowledge distillation.

**Audience:**

Yes

**Claims And Evidence:**

Yes

**Requested Changes:**

Please address my concerns mentioned above.

**Strengths And Weaknesses:**

Strengths:
1.	The authors carefully examine the mathematical formulation and process in the original paper, and correct the error.
2.	To find out the genuine factor for the performance improvement, the authors carry out detailed and sufficient ablation experiments and visual analysis.
Weaknesses:
1. The differences between MRL and other baselines are not clearly stated, especially for CLDRD. As the authors mainly follow the settings of CLDRD and regard it as an important competitive method, the similarities and differences between MRL and CLDRD should be introduced in detail.
2. Some notations are not explained. For example, the rtqd in the page 7 in the un-numbered equation is not explained. The authors should check through the paper because there are many easily-confusing notations and hyperparameters.

---

### Review · Reviewer_vDe3 · 2024-05-17

**Summary Of Contributions:**

1.	The authors identified and fixed a mathematical error in Multivariate Representation Learning (MRL).
2.	The paper provided critical information necessary to reproduce MRL performance, which was not written in the original paper.
3.	Extensive ablation studies conclude that model initialization and knowledge distillation from the re-ranker are crucial for training a retriever that generates MRL outputs.
4.	The proposed approach enhanced performance while reducing the hyperparameter search space by employing log-variance predictions instead of Softplus activation.

**Audience:**

No

**Broader Impact Concerns:**

There are no broader impact concerns.

**Claims And Evidence:**

No

**Requested Changes:**

To validate the findings demonstrated by the authors through ablation studies, it is necessary to present results from experiments conducted with a batch size of 512.

**Strengths And Weaknesses:**

Strengths:

•	The reliability of the experiments was increased by releasing the code publicly and conducting numerous experiments.
•	The use of tables and figures facilitated a clearer understanding of the content.
•	Minor mathematical errors were identified and fixed.

Weaknesses:

•	The batch size significantly influences the performance of models trained through contrastive learning methodologies utilizing positive and negative samples. Although the original MRL paper used a batch size of 512, this paper reduced it to 15 for fairness compared to other baselines. Consequently, the ablation study results are thus based on this reduced batch size. This paper claims that MRL performs poorly in downstream performance compared to other baselines and fails to capture uncertainty. However, these conclusions lack credibility as they are derived from models trained with drastically reduced batch sizes. An ablation study with progressively increased batch sizes for MRL and baseline models is necessary to validate these claims credibly.
•	The paper needs to thoroughly address the previous studies. For example, the claim that "existing dense representations cannot capture or express uncertainty" must be substantiated with references.

Minor Weaknesses:

•	It would be beneficial to explain the acronyms when initially referencing "IR," which appears to stand for Information Retrieval.
•	The paper often lacks explanations of specific terms. For instance, explaining the purpose of dense retrieval and the concept of curriculum learning would enhance clarity and comprehension.

---

### Review · Reviewer_iL1v · 2024-06-06

**Summary Of Contributions:**

This paper attempts to reproduce the main results of [1] and [2] at the task of dense information retrieval from queries. The models proposed in all three papers ([1],[2] and the present submission).

 The main original contribution in [2] (the CL-DRD method) is the application of curriculum learning with knowledge distillation to retrieval tasks: an inefficient teacher model is first trained, and the student model relearns the ranking function based on supervised information provided by the teacher model in batches of increasing difficulty via the knowledge distillation loss from equation (12) in [1] which is equation (2) in [3] and Equation (13) in the submitted paper.  The construction of the training set for each paper differs: in [1], the positives are taken from the ground truth labels and the negatives are obtained by applying another method (“BM 25 with an asynchronously updated ANN index”), and it appears the $y_t$s are taken to be the scores produced by the teacher model; on the other hand, in both  the earlier work of [2] and the present paper, the positive samples are produced by the teacher model and the $y_t$s can only take three values corresponding to “positive”, ‘negative” and “hard negative”, all produced by the teacher model. (In the present paper, the positive examples are also weighted in a more refined way by relevance as ascribed by the teacher model (cf. equation on page 7).

The main additional contribution in [1] consist in what both [1] and the present submission refer to as “multivariate representation learning”, which means that instead of a feature vector, each query and document is attributed a multivariate normal distribution. The loss function is the changed from cosine similarity (or something  similar) to the KL divergence between both distributions. It is also shown that this new similarity measure can be expressed as an inner product between feature vectors constructed from the statistics (mean and variance) which define the multivariate distributions being learned (see equations (11) and (12) in the present paper. In addition, a third important component of the model is the initialisation of the model with [3].  Furthermore, the original paper [1] also studies the claimed correlation between the variance of the learned distribution and the difficulty of the retrieval task (Query Performance Prediction, QPP).

The aim of this paper is to reproduce the results in [1] thoroughly and provide algorithmic modifications which improve performance. In particular, the main contribution could be interpreted as the highlighting of an error in the original paper in the calculation of the KL divergence: the original authors incorrectly calculated the term $\text{tr}(\Sigma^{-1}_{D} \Sigma_Q)$. In addition, as mentioned above, the authors of the present paper propose a different version of the distillation strategy which is much closer to the original work of [2].  The experiments claim to demonstrate that the original formulation of [1] is non-reproducible and leads to very poor results which are easily beaten by trivial baselines and far away from the reported results. In addition, the authors study the modified version of the slightly algorithm which they propose and show that it achieves strong performance, but demonstrate through an ablation study that the performance gains are almost entirely due to the knowledge distillation component, which is consistent with the observation that the [2] baseline is the top performing one in many experiments. In particular, they conclude that [2] is reproducible.  In addition, like the original paper [1], the authors study the correlation between (minus) the determinant of the covariance matrix and the performance through Query Performance Prediction experiments and conclude that there is a small positive correlation between the constructed quantities, though the results are not consistent. The authors also point out that the original papers authors claim to have used the determinant itself (without a minus sign), but that is disputable. Finally, experiments are performed comparing the variances of clean and noisy queries (the noisy queries involve typographical errors), yielding the conclusion that the covariance determinant is actually positively correlated with the clean-ness of the data, which runs counter to the intuition of the model provided in [1]. The authors conclude that the model uses the covariance matrix as a predictor of relevance rather than a measure of uncertainty.




============




References:


[1] Hamed Zamani, Michael Bendersky. “Multivariate Representation Learning for Information Retrieval “. SIGIR 2023. [Paper to reproduce]


[2] Hansi Zeng, Hamed Zamani, Vishwa Vinay. “Curriculum Learning for Dense Retrieval Distillation”. SIGIR 2022.  [Earlier paper on which [1] is based]


[3] Sebastian Hofstätter, Sheng-Chieh Lin, Jheng-Hong Yang, Jimmy Lin, and Allan Hanbury.  Efficiently teaching an effective dense retriever with balanced topic aware sampling.  SIGIR 2021.  [Baseline and Pretraining for [2,3,present paper]]

**Audience:**

Yes

**Broader Impact Concerns:**

1. I find the entire enterprise of claiming another paper is irreproducible quite challenging ethically: personally, I would be hesitant to ever use the word “mistake” or “error” to describe even an indisputable error in another paper as such, regardless of the validity of the statement. I am quite surprised to see the authors of the present paper so willing to take this risk. It is probably possible to phrase such things more tactfully.

 2. How much of current applied research is really reproducible and really provides statistically significant and robust performance improvements over all existing contemporary work? Is it fair to call out a subset of published research for failing to achieve that improbably achievable aim? I concede that this paper is potentially “more unreproducible than average” due the mathematical error (such serious errors are less common than expected, despite the obvious lack of a truly effective peer review system), but I don’t think that the authors of [1] are guilty beyond reasonable doubt of academic dishonesty. It seems more likely there was some degree of confusion during the implementation.

**Claims And Evidence:**

Yes

**Requested Changes:**

1. Try to express the perceived imperfections of the original paper more tactfully.
2. Run the full range of hyper parameter search as in the original paper, including the learning rate and if possible, the batch size.
3. Describe the learning setting in more detail, including some modifications such as LogVar (section 5.4 should be rewritten). Regarding the general learning setting, I think more details about the tokenisation and input format should be welcome. In particular, the original paper contains the statement (page 4) “ For example, we convert an input query “ Neural information retrieval” to “[CLS] [VAR] neural information retrieval”. I can’t really make sense of it as I am not from this field. It would be nice to explain to enhance  the reader friendliness and broaden the audience.

**Strengths And Weaknesses:**

Strengths:

1.  It must be said that there is indubitably a mathematical error in the calculation of the KL divergence in [1], and it is correctly pointed out and corrected in the present paper.
2. The experimental evaluation is relatively thorough, to a degree which is higher than average for machine learning papers.
3. Although I feel uneasy about this, I do feel like the results are *highly credible*. In particular, the fact that the covariance matrix doesn’t correlate with uncertainty and that the uncertainty predicting module is essentially nothing but an additional architectural trick with no relationship with the proposed Bayesian interpretation is very highly believable to me, since the loss function only uses the ranking performance as a training signal.  However, **I didn’t look at the code**, so I cannot ultimately make a definitive judgement.
4. The paper is generally very well written.


Weaknesses:

1. The statements against [1] are quite harsh. Here, I feel the authors go further than that: not only  do they describe errors in the original paper using very frank language, they also use similarly strong language in places where it doesn’t even pass the test of indisputable correctness: for instance, on page 7, the authors describe their modification of the knowledge distillation strategy by saying that the original strategy “contradicts training with knowledge distillation from various points of view”. At the end of the day, although I tend to generally agree with the authors of the current paper about the statement, I don’t feel it is objective enough to warrant such harsh words.
2. It seems that when the authors of the present paper claim that the results in [1] are irreproducible and that the method in [1] doesn’t actually perform that well, what they really mean is that there may be inconsistencies between the implementation of [1] and the description in the main paper of [1]. For instance, it is not clear to me whether it might be the case that the authors of the original paper [1] actually implemented the correct version of the KL divergence and only wrote it wrong in the text. In addition, a **key component of the controversy** appears to be the choice of batch size: the authors of the present paper claim that a **batch size** of 15 is the maximum they can fit into memory, whilst the results in [1] quote 512. This seems to explain  much of the observed performance differences. Is it possible that the inability to fit the batch into memory is due to inefficient representation in the implementation of the current paper? It is also not absolutely clear beyond reasonable doubt whether the original paper’s authors might have used similarly high batch sizes for every method (including the baselines): if that were the case, that would still make the original results valid. In any case, the position of the batch size as a key potential factor isn’t duly represented in the presentation of the paper.
3. The range of search for the learning rate  is not as thorough as in the original paper [1]. The experiments must be rerun with the same configuration ( $1\times 10^{-6}$ to $1\times 10^{-5}$ in additive increments of $1\times 10^{-6}$).
4. Some of the explanations could be clearer for readers not familiar to this field. In particular, I would enjoy a more general introduction to retrieval as a task in general (input, output) and a description of the architectures. This is particularly important for this reproducibility study since the architecture is not described in much detail in either paper.
5. One of the modifications of the model which is proposed in the present paper is called “LogVar” and is only described as “predicting the log covariance instead”. This sentence fragment is the whole description of this new method… This is obviously irreproducible and should be substantially expanded.

---

> ### Author Response · Authors · 2024-06-19
> **Response to reviewer iL1v**
>
> Thank you for the very thorough review and comments. We address your comments and feedback below.
>
> - Sign of the determinant: we assumed that the original paper uses the determinant without a minus sign based on the following statement in the last page of the paper – “This observation motivated us to use the learned $|\Sigma_Q|$  for each query as a pre-retrieval query performance predictor (QPP).”.  Indeed, the authors might have meant ‘a function of the variance’, and might have included a minus. We will amend the language in Section 4.4 / page 8 to reflect this.
>
> - On the language used in page 7 about KD: We agree that the language used in this section is perhaps too strong. We will attempt to address this in a more nuanced manner in the revised version.
>
> - We agree with the reviewer’s comment that the incorrect KL derivation is a typographical error, rather than an incorrect implementation. Indeed, in Table 3, we show that using the original reported formula results in very poor performance, which makes it very likely that the error was only in the paper and not in the author's implementation. We will amend the paper to reflect this assumption.
>
> - On the batch size: While it is entirely possible that the authors used a large batch size for the baselines, we argue that this is unlikely –  the numbers for CL-DRD, for instance,  in the MRL paper and the original CL-DRD paper are identical. It is therefore likely that a batch-size of 8 was used in the implementation of  CL-DRD.
>
> - Our experiments are designed with a fair comparison in mind.  That is, all of our baselines use a similar training setup (to the extent that this is possible), with the same batch size i.e., reproducibility, not replicability [C]. We argue that the comparison of models with a batch size of 8 and 512 is unfair, given that there is evidence that batch size impacts retrieval performance [A]. Furthermore, we also argue that reproducibility is primarily concerned with generalization to different settings [C, D], and our current setup is justified.
>
> - On larger batch-size experiments: While we planned to train the model (as-is) on multiple GPUs to accommodate a batch size of 512, we quickly abandoned this as it was not feasible with our current resources. For a batch-size of 512, we would require 32x40GB GPUs, which is (a) not available to us in our cluster (b) prohibitively expensive (1K USD per day).
> We re-implemented the model using grad-cache [A], allowing us to obtain a batch size of 512. At the moment we have some preliminary results on MRL-Orig. (MRL is following the training setup as presented in the original study). We observe that while training with a 512 batch improves over the performance achieved with a batch of 15, it still does not achieve the reported results, nor does it beat the baseline with a much smaller batch size. Moreover, we are actively conducting further experiments for various batch sizes, including 32, 64, 128, 256, and 512. The same experiments will be conducted for MR-Ours as well.
>
> We will also take the reviewer’s feedback into account and amend the paper to highlight that (a) batch size is an important factor and (b) the experimental setup is not identical to MRL, but our intention is to have a fair comparison and test generalization to lower batch sizes.
> |            | Model     | Batch | MS MARCO | TREC-DL'19 | TREC-DL'20 | SciFact | FiQA    | TREC-COVID | CQADupStack |
> |------------|-----------|-------|----------|------------|------------|---------|---------|------------|-------------|
> |            |           |       | MRR@10   | NDCG@10    | NDCG@10    | NDCG@10 | NDCG@10 | NDCG@10    | NDCG@10     |
> | Reproduced | MRL-Orig. | 15    | .255     | .576       | .534       | .305    | .146    | .169       | .185        |
> | Reproduced | MRL-Orig. | 512   | .297     | .667       | .63        | .459    | .213    | .413       | .267        |
> | Reproduced | MRL-Ours  | 15    | .375     | .721       | .667       | .605    | .293    | .510       | .320        |
> | Reported   | MRL       | 512   | .393     | .738       | .701       | .683    | .371    | .668       | .341        |
>
> - On re-running with larger hyperparameter search space: Due to limited resources on our cluster, we chose to have a (slightly) smaller search space for MRL. However, we note that all of our baselines also had the same search space, making the comparison fair.

---

> > ### Author Response · Authors · 2024-06-19
> > **Response to reviewer iL1v**
> >
> > - On the possibility of inefficient/incorrect implementation: we used a widely popular IR framework for our implementation (Tevatron).  With Tevatron (without modification), we can accommodate a batch-size of 15 for the DPR model on our GPUs. While we cannot rule out that our implementation is incorrect / inefficient, we think it unlikely that our implementation is inefficient. Or, perhaps more accurately, our implementation is at least as efficient as tevatron. We also did our best effort in checking our implementation with multiple rounds of code review to reduce the possibility of an incorrect implementation. We welcome the reviewers to review the code submitted with the paper (in particular – `src/tevatron/modeling/dense_mvrl.py`).
> >
> > - LogVar: we will expand this section significantly. The main intent was to use an alternative to ensure that the co-variance remains positive semi-definite. One way to ensure positivity  is to use a softplus activation, which is the approach taken  in the original paper. Predicting the log-variance instead of using the softplus – and exponentiating the log-variance when needed –  would also ensure positivity, with the advantage of not needing a hyperparameter search for $\beta$.
> >
> > - We will include a brief introduction to retrieval in the revised version, including the changes mentioned in the Requested Changes #3.
> >
> > - Finally, we do not, directly or indirectly, imply that any author is guilty of academic dishonesty. Instead, we are trying to communicate that we could not obtain the reported results with the information provided by the original paper; a result that is not uncommon in ML research. There has been considerable research in several fields showing that a significant portion of research is irreproducible [B, D, E], making reproducibility studies an important way to test the validity of ideas and methods. That said, we hold the research contributions of the original paper in high regard (as we emphasize in the Conclusion section of our work), particularly the core idea at the heart of the MRL paper, which led to this study.  We work with uncertainty estimation a lot and, in fact, this work would have been very useful to us, if we'd managed to reproduce it. Our motivation was really to test the validity of its core idea, and report it as objectively as possible.
> >
> > [A] Gao, Luyu, et al. "Scaling deep contrastive learning batch size under memory limited setup." arXiv preprint arXiv:2101.06983 (2021).
> > [B] Baker. 2016. Is there a reproducibility crisis? Nature.
> > [C] ACM. 2018. Artifact Review and Badging.
> > [D] Pineau et al. 2020. Improving Reproducibility in ML Research
> > [E] Hutson. 2018. Artificial intelligence faces reproducibility crisis. Science

---

### Author Response · Authors · 2024-06-29
**Revision**

We have uploaded an updated  version of the paper with the following revisions:
- Made requested changes about the sign of the determinant (Section 5.2.1 and 4.4)
- KL typographical error (Introduction,  Section 3.1)
- Expanded log-var section (Section 5.4)
- Additional details about pre-processing, tokenization and obtaining the variance (Section 3.0)
- Added a table that highlights the differences between CLDRD and MRL (Section 4.3, 5.1 and 5.3)
- Significantly edited Section   4.4 and  5.1.
- Added missing description of $r^t_{qd}$.

Furthermore, we have obtained partial results using a batch size of 512 for MRL-Orig as reported earlier, and are running additional experiments for MRL-Ours and CL-DRD. In addition to the batch size of 512, we are experimenting with 32,64,128 and 256. We have not included these numbers in the revised version of the paper since we are awaiting the results for some models / batch sizes.  We intend to update the paper again once these results are obtained (by 10 July at most).

---

### Decision · Action_Editor_M1kh · 2024-08-05

**Recommendation:** Reject

**Comment:**

Per the above, we would encourage the authors to re-submit a revised version of the paper which includes further results in the large batch setting. It is of interest to see whether the main trends observed in the small batch setting still hold, and whether the MRL results are overall more favourable.

Comments:
- "or risk" -> what is this risk?
- "the retrieval system has minimal to no prior knowledge about the distribution of the queries" -> but it may be trained on a large body of queries? Also, seems at odds with the subsequent sentence claim that "Unlike queries, the retrieval model has prior knowledge..."
- "MRL treats both the mean and the variance as point estimates" -> not clear what this means at this stage. We typically view parameters as being point estimates as opposed to distributions.
- Equation 1, perhaps the LHS should be $( \mu_Q, \Sigma_Q )$; similarly for Equation 2.
- Equation 8, consider stating the result as a Proposition or similar, with the derivation in Equations 5-7 serving as the proof.
- Equation 9, $\cdot$ is unnecessary
- pg 6, "That said" appears awkward; omit?
- Equation 13, use \big\left| and \big\right|
- pg 7, use \citep{.} for Nguyen et al., 2016
- pg 7, definition of $y^t_q(d)$, in the RHS, use $d$ rather than d.
- pg 9, consistent use of \tt when referring to CLS and VAR tokens.
- pg 13, $\Sigma_{clean}$ -> $\Sigma_{\rm clean}$; similarly for $\Sigma_{corrupted}$
- pg 15, definition of softplus, replace * with $\cdot$

**Audience:**

The findings in the paper would likely be of interest to practitioners in IR, as it concerns the viability of techniques that capture uncertainty.

**Claims And Evidence:**

The paper presents a reproducibility study of the multivariate representation learning (MRL) framework from Zamani & Bendersky (2023). The paper's main claims are:
- the original MRL paper contained a typographic error in the derivation of a certain KL divergence
- certain important design choices were not fully specified in the original MRL paper, and with various reasonable imputations for these choices, the likely candidate for the reported gains are owing to the distillation loss

The first claim was accepted by all reviewers.

For the second claim, it was argued by reviewers that the choice of batch size may prove very important, and that this point may not be fully represented in the paper. The MRL paper used a batch size of 512, while the present paper reports results for a batch size of 15 owing to computational constraints. Reviewers expressed concern that it is not clear whether some of the observations in the present paper would still hold in the large batch setting. In response to this:
- the authors note that the other baselines typically employ small batch sizes, which motivates a fair comparison against a small batch MRL. This point is valid. Some reviewers nonetheless expressed interest in understanding whether the issues with MRL are manifest in the large batch setting.

- The authors included batch size 512 results for the _original_ MRL distillation recipe. The authors note that their proposed version of MRL, which is closer to CLDRD, does not rely on in-batch negatives, and so may not benefit significantly from a larger batch size. While this argument has some merit, given that it is a deviation from the setting of the original paper, reviewers were keen on concrete empirical support. The authors intended to produce results for the proposed MRL method in the large batch setting as well; unfortunately, similar results for the proposed distillation recipe could not be completed over the course of the review period.

Given this, one possibility might be for the authors to re-frame their paper as specifically discussing the low batch size setting. However, a conclusive analysis of the MRL method in a similar setting to the original paper would be ideal for a replication study.

Separately, another critique was that the original paper's wording when discussing MRL may have been overly harsh. Following the discussion, the authors revised the paper to adopt a more neutral tone in describing the MRL paper. Nonetheless, from my reading, I would also suggest revisiting the following:
- Abstract, "[MRL] does not provide all of the necessary information to facilitate reproducibility" -> could simply say "does not fully specify certain important design choices that can strongly influence performance"
- Abstract, "We further contribute a thorough ablation study which is absent from the original paper" -> could simply say "We expand on the results from the original paper with a thorough ablation study"
- pg 2, "the fact that neither the source code nor model checkpoints are released" -> could simply say "Owing to the source code and models being unavailable"
- pg 16, "Lack of details" -> could simply say "(Incomplete?) Specification of design choices"

**Resubmission Of Major Revision:**

The authors may consider submitting a major revision at a later time.